# The Theoretical and Experimental Investigation of the Fluorinated Palladium β-Diketonate Derivatives: Structure and Physicochemical Properties

**DOI:** 10.3390/molecules27072207

**Published:** 2022-03-28

**Authors:** Svetlana I. Dorovskikh, Denis E. Tryakhov, Darya D. Klyamer, Alexander S. Sukhikh, Irina V. Mirzaeva, Natalia B. Morozova, Tamara V. Basova

**Affiliations:** 1Nikolaev Institute of Inorganic Chemistry, Siberian Branch of Russian Academy of Sciences, 3 Acad. Lavrentiev Ave., 630090 Novosibirsk, Russia; klyamer@niic.nsc.ru (D.D.K.); a_sukhikh@niic.nsc.ru (A.S.S.); dairdre@gmail.com (I.V.M.); mor@niic.nsc.ru (N.B.M.); basova@niic.nsc.ru (T.V.B.); 2Chemistry Department, Materials Science Faculty, Novosibirsk State University, 2 Pirogova Str., 630090 Novosibirsk, Russia; tryakhovd@gmail.com

**Keywords:** palladium(II) β-diketonate derivatives, crystal structure, Hirshfeld surface analysis, DFT calculations, TG study

## Abstract

To search for new suitable Pd precursors for MOCVD/ALD processes, the extended series of fluorinated palladium complexes [Pd(CH_3_CXCHCO(R))_2_] with β-diketone [tfa−1,1,1-trifluoro-2,4-pentanedionato (**1**); pfpa−5,5,6,6,6-pentafluoro-2,4-hexanedionato (**3**); hfba−5,5,6,6,7,7,7-heptafluoro-2,4-heptanedionato (**5**)] and β-iminoketone [i-tfa−1,1,1-trifluoro-2-imino-4-pentanonato (**2**); i-pfpa−5,5,6,6,6-pentafluoro-2-imino-4-hexanonato (**4**); i-hfba-5,5,6,6,7,7,7-heptafluoro-2-imino-4-heptanonato (**6**)] ligands were synthesized with 70–80% yields and characterized by a set of experimental (SXRD, XRD, IR, NMR spectroscopy, TG) and theoretical (DFT, Hirshfeld surface analysis) methods. Solutions of Pd β-diketonates contained both *cis* and *trans* isomers, while only *trans* isomers were detected in the solutions of Pd β-iminoketonates. The molecules **2**–**6** and new polymorphs of complexes **3** and **5** were arranged preferentially in stacks, and the distance between molecules in the stack generally increased with elongation of the fluorine chain in ligands. The H…F contacts were the main ones involved in the formation of packages of molecules **1**–**2**, and C…F, F…F, NH…F contacts appeared in the structures of complexes **4**–**6**. The stability of complexes and their polymorphs in the crystal phases were estimated from DFT calculations. The TG data showed that the volatility differences between Pd β-iminoketonates and Pd β-diketonates were minimized with the elongation of the fluorine chain in the ligands.

## 1. Introduction

Palladium-based nanomaterials are an integral part of hydrogen energy devices [1,2], and they are in demand in catalysis [3], the chemical industry [4,5], and medicine [6]. The latest trend in the field of non-invasive diagnostics is associated with the study of exhaled breath to detect gaseous biomarkers of respiratory (NO, NO_2_), kidney (NH_3_), and other diseases [7,8,9]. This has stimulated interest in new sensor materials, particularly those based on semiconductors decorated with Pd-containing particles, as well as in the methods of their fabrication [10,11,12].

The advantages of using gas-phase methods such as physical vapor deposition (PVD), metal-organic chemical vapor deposition (MOCVD), or atomic layer deposition (ALD) for the deposition of various palladium materials from ultrafine particles to thick coatings, using various types of substrates (semiconductors, isolators, metals, glasses, and others), have already been discussed in the literature [13,14,15,16]. The implementation of chemical deposition methods (MOCVD, ALD) involves the use of a palladium precursor that should have high volatility, thermal stability in the condensed phase during MOCVD/ALD, and storage stability [14,16]. To date, palladium(II) MOCVD/ALD precursors have consisted of compounds with organic ligands such as aminoalkoxides [17], β-diketonates [16,18,19,20], β-iminoketonates [18,21,22], and Schiff base [23]. Among the above compounds, Pd β-diketones and their derivatives are interesting objects for studying the effect of the ligand structure on the structure and physicochemical characteristics of complexes [17,18,21]. According to several studies [19,21,22,24], the introduction of CF_3_ groups to the ligand gives the corresponding palladium complexes high volatility and good thermal stability. To date, the effect of fluorosubstitution in the ligand on the structure and physical properties of Pd β-diketonate derivatives has been discussed only for a limited number of compounds: [Pd(tfa)_2_], [Pd(hfa)_2_], [Pd(i-tfa)_2_], [Pd(Mei-tfa)_2_], [Pd(dmht)_2_] (tfa–MeCOCHCOCF_3_, hfa–CF_3_COCHCOCF_3_, i-tfa–MeCNHCHCOCF_3_, Mei-tfa–MeCNMeCHCOCF_3_, dmht–MeCNNMeCHCOCF_3_), compared with their non-fluorinated analogues [19,21,22,24]. The effect of the degree of ligand fluorination on the compound properties also remains insufficiently investigated. 

The use of theoretical approaches based on DFT calculations in combination with single-crystal X-ray diffraction (SXRD) and IR and NMR spectroscopies provides additional opportunities for studying the energies of crystal lattices and individual molecules of the complexes [25,26], for estimation of the strength of metal–ligand bonds [27,28] and the distribution of tautomeric forms [29]. From a practical point of view, spectral methods, together with DFT, are widely used to estimate kinetic parameters and probabilities of the cleavage of bonds in the precursor molecules [30] and to predict the decomposition pathway of the precursor vapors on various substrates during the MOCVD/ALD process [31]. 

Thus, the aim of this work is to further develop the chemistry of fluorinated palladium β-diketonate derivatives using a combination of experimental (SXRD, IR, NMR, TG) and theoretical (DFT, Hirshfeld surface analysis) methods. A series of Pd complexes with the general formula [Pd(CH_3_CXCHCO(R))_2_] (Table 1) were synthesized and isolated with a high yield (70–80%) to study the effect of the length of the fluorinated substituent chains and the combinations of donor atoms in the ligand on the structure and physicochemical properties of the Pd complexes.

## 2. Results and Discussion

### 2.1. Characterization of Complexes **1**–**6**

A series of complexes **1**–**6** were obtained with high yields, according to Figure 1.

To prevent undesired hydrolysis of the ligand, dry methanol was used as a solvent. The method of zone vacuum sublimation at P = 10^−2^ Torr and t = 60–180 °C was used to isolate an individual phase of each complex (Table 1). All the resulting compounds sublimed almost completely and were yellow-orange powders, stable in air. The complexes required storage at cool temperatures due to their high volatility. The elongation of the fluorine chain in L_F_ led to a decrease in the melting points of **3**–**6** vs. **1**–**2**.

All major peaks in the ^1^H and ^13^C NMR spectra of the compound solutions in CDCl_3_ could be easily assigned. Although the signals of the –NH groups of **2**, **4**, and **6** overlap with the ^1^H signals corresponding to the solvent residual, they have different widths and are quite distinguishable (Appendix A).

For **1**, **3**, and **5,** the signals in both the ^1^H and ^13^C spectra are doubled (Appendix A). This indicates that the solutions appear to contain both *cis* and *trans* isomers of the complexes. Theoretically, such isomers are possible for all the complexes under study (Figure 1). 

We performed DFT calculations to find out which isomer was energetically preferable. For all complexes, *trans* isomers had lower energies (Table 2). However, Pd complexes with β-iminoketone ligands showed a larger difference in energy between isomers than complexes with β-diketone ligands. This is consistent with the fact that solutions of **4** and **6** contain only one isomer. The ^1^H NMR spectrum of the solution of **2** contained an additional set of small signals, (Appendix A), which probably indicates a small admixture of the *cis* isomer. Probably, this occurs because **2** has the smallest ligands, which may lead to the lowest barrier for *cis–trans* transition among the complexes with β-iminoketone ligands. In the ^13^C NMR spectrum of **2**, we could not see this additional set of signals, because of the lower signal-to-noise ratio. Another interesting feature of **2** is that its ^1^H NMR spectrum shows the splitting of each line into a doublet (Appendix A), which may be attributed to the indirect spin–spin coupling of the –CH_3_ group and γ-H to the –NH group.

The IR spectra of complexes **1** and **2** have already been described in our previous paper [30]. The experimental IR spectra of complexes **3**–**6** are presented in Figure 2. 

The most intense bands in the experimental and calculated spectra of complexes **3**–**6** are related to both stretching and deformational vibrations of fluorinated groups and were located in the range of 1100–1200 cm^−1^. The stretching vibrations of C=C, C=O, and C=N groups could be observed as intense, closely lying bands in the range of 1500–1600 cm^−1^. According to the DFT calculations, most of the peaks in the range from 500 to 1000 cm^−1^ corresponded to the complex group deformations of the chelate rings. Most of these vibrations were mixed with the C-H vibrations. Bands with a high contribution of Pd–O and Pd–N stretching vibrations were observed in the range of 400–700 cm^−1^. The IR spectra of β-iminoketonate complexes **4** and **6** were characterized by the presence of a strong band at 3340–3360 cm^−1^, related to ν(N–H) vibrations. 

### 2.2. Single-Crystal and Powder XRD Analyses

Crystals of **2**–**6** suitable for the SXRD study were grown by zone sublimation. These crystals were labeled **2, 3_1, 4**, and **6**. In the case of **5**, two types of crystals designated **5_1** and **5_2** were isolated during zone sublimation (see details below). All the above-mentioned crystals consisted only of *trans* isomer molecules. Our attempts to obtain *cis* isomers of **3, 5** by evaporation of their chloroform solutions were unsuccessful. However, this approach led to the isolation of new crystal modifications (*trans* isomers) of **3** and **5** named **3_2** and **5_3**, respectively. The unit cell parameters and refinement statistics for all the studied crystals are summarized in Table 3.

Figure 3 shows the molecular packing of **3**–**6**. The crystal structures of **1** and **2** have already been described in the literature [32,33]. However, since the structure of **2** in CSD does not contain atomic coordinates for the hydrogen atoms, which are necessary for the proper analysis of Hirshfeld surfaces, we collected a new *hkl* dataset and redetermined the crystal structure by specifying the coordinates of the hydrogen atoms. 

From the point of view of the molecular structure, all the compounds **1**–**6** can be considered as two metallocycles that have a common central Pd atom, with a flat part of the molecule in the shape of a figure eight and two perfluoroalkyl fragments attached to the periphery of the metallocycles. Since the molecules are flat, they tend to be packed in stacks, arranged either uniformly, as in case **1**, or in a herringbone pattern, as in case **2**. However, this packing motif is distorted as the peripheral perfluoroalkyl fragments become bulkier. 

The two polymorphs of **3** have the following characteristics: polymorph **3_1** crystallizes in a triclinic unit cell with Z = 1, and polymorph **3_2** crystallizes in a monoclinic unit cell with Z = 4. 

At first glance, polymorph **3_1** has a similar style of molecular packing and similar unit cell parameters as a crystal of **1**, where the molecules are arranged in uniform stacks with a distance between the molecules in the stack of 4.79 Å (the distance between the planes of the square formed by the atoms around Pd) and a packing angle (the angle between the above-mentioned plane of the square and the stacking direction) of 45.57°. The molecules in **3_1** are shifted relative to each other, so that their metallocycles do not actually overlap. As a result, the molecular packing of **3_1** can better be described as layered, with a distance between the molecules in adjacent layers of 4.90 Å. 

The polymorph **3_2** also contains molecules packed in stacks, but the structure of the stacks is noticeably distorted. The molecules are no longer parallel to each other (the angle between adjacent molecules is 10.6°) and the position of –C_2_F_5_ groups alternates between top right/bottom left and top left/bottom right when viewed along the packing direction. The polymorph **3_2** has a noticeably lower density of 415.4 Å^3^ per molecule compared to the polymorph **3_1**, which has a density of 379.96 Å^3^ per molecule. The distance between the molecules and the packing angle are 3.357–3.440 Å and 42.732° for the polymorph **3_2**.

The polymorph **5_2** has two symmetrically independent molecules within the unit cell; the –C_3_F_7_ group in one of the molecules is disordered over two positions with a ratio of 0.63:0.37. When viewed along the *a*-axis, it seems that these molecules represent two types of molecular stacks (each stack consists of only one type of molecule), arranged in a herringbone pattern. However, the central metallocycles of **5** overlap in only one type of stack. The distance between the molecules in this stack is 3.439 Å, and the stacking angle is 53.35°. The distance between the molecules in the second type of stack is 3.908 Å. 

The polymorph **5_3** has a molecular packing style similar to the polymorph **3_2,** with molecules packed in stacks in such a way that the position of the bulky –C_3_F_7_ groups alternates inside the stack. The distance between the molecules in the stack is 3.469 Å, the stacking angle is ~41.8°, and the angle between the molecules within the stack is 11.71°. However, unlike in complex **3**, the molecular stacks in polymorph **5_3** are more uniform, with Pd atoms lying strictly on the same line, while Pd atoms in polymorph **3_2** deviate from the mean squared line by 0.28 Å.

Finally, the polymorph **5_1** also contains molecules packed in stacks with the positions of –C_3_F_7_ groups alternating within the stack. The distance between the molecules inside the stack is 3.414 Å, the stacking angle is 40.58°, and the angle between the molecules in the stacks is 10.15°. The polymorphs **5_1** and **5_3** have a similar structure of molecular stacks, but the arrangement of stacks is different. In **5_3,** the molecular stacks are arranged in the form of a parquet motif, while a hexagonal motif is clearly visible in **5_2**.

Unlike for their β-diketonate analogues, no polymorphism was observed for **4** and **6**. Complex **4** has the same packing style as the polymorph **5_2,** with a prominent hexagonal motif in the arrangement of molecular stacks. The distance between the molecules in **4** is 3.394 Å, the angle between the molecules within the stack is 0.29° and the stacking angle is ~6.4°, which is significantly less than in other β-diketonate complexes with a similar packing style, namely **3_2** and **5_2**. Finally, **6** crystallizes in a P-1 space group with Z = 1 and has the same molecular packing style as the polymorph **5_1,** with molecules arranged in uniform stacks. The distance between the molecules in the stack is 3.397 Å and the stacking angle is 47.59°.

Atomic bond lengths, angles, and other parameters related to the geometry and molecular packing of Pd β-diketonate and β-iminoketonate complexes are shown in Table 4. 

In summary, we can say that all complexes have a similar Pd coordination sphere. As for the molecular packaging, differences are observed for complexes **3_1** and **4**, in that **3_1** is the only compound whose molecules are arranged in layers rather than stacks, and **4** is characterized by a relatively small stacking angle of 6.4° compared to the other compounds.

Theoretical diffractograms of the complexes (Figure 4, black and blue lines), which were obtained based on the SXRD data of the complexes **1**–**6** and their polymorphs, were used to identify powders of the as-sublimed complexes **1**–**6** (Figure 4, red lines). Most of the complexes **1**–**6** are single-phase complexes. Some typical diffractograms for complexes **3**–**6** are shown in Figure 4. Only the modification **3_1** was detected in the powder of the sublimed complex **3** (Figure 4a). In case of complex **5**, we succeed in isolating the individual phases of **5_1** and **5_2**. The **5_1** modification formed during sublimation of complex **5** at a temperatures of 60–65 °C (Figure 4c). An increase in the sublimation temperature to 80–85 °C led to the formation of a mixture of modifications **5_1** and **5_2** (Figure 4d). The modification **5_1** still remained the main phase in the as-sublimed complex **5**, while **5_2** accumulated in the powder of complex **5** when heated. Only one modification, **5_2,** was formed during sublimation of complex **5** at 100–110 °C (Figure 4e).

### 2.3. Hirshfeld Surface Analysis of Complexes **1**–**6**

The analysis of Hirshfeld surfaces (HSs) was used to visualize the main intermolecular contacts, in order the reveal their correlation with the volatility of the compounds and the calculated crystal lattice energy data. The Hirshfeld surface analysis was not carried out for complexes **3_2** and **5_3,** because their powders were not single-phase.

Figure 5 shows HSs for Pd β-diketonates, mapped with the d_norm_ property (range −0.18 to 1.2). Mapping using d_norm_ makes it possible to visualize all intermolecular contacts quickly and simply and shows all close contacts in the form of red dots (the more intense the red color, the stronger the intermolecular contact). For ease of comparison, the d_norm_ range was the same (from −0.18 to 1.2) for all Hirshfeld surfaces. The lower limit (−0.18) was chosen deliberately since the closest intermolecular contact observed among all Pd complexes was −0.175 Å (the distance between two atoms in neighboring molecules was 0.35 Å, which is less than the sum of their van der Waals radii). The minimal d_norm_ value is shown at the bottom of each HS (Figure 5).

The HS of complex **1** has a pair of weak 2.454 Å F…H contacts from the edge of the molecule and one 2.490 Å F…H contact on its front side. The HS of the complex **3**_**1** features a pair of 2.831 Å F…F close contacts, one 2.868 Å F…F contact, and no F…H contacts. The polymorph **5_2** contains two independent molecules in the unit cell, and the second molecule has its –C_3_F_7_ groups disordered over two positions with a ratio of 0.63:0.37. For this reason, it is necessary to consider three HSs: one for the first molecule and two disorder variants for the second molecule. Only one HS for the first molecule is considered because it varies very little depending on the chosen disorder variant of the second molecule. The HS of the first molecule has one relatively prominent 3.061 Å C…F contact and a pair of 2.853 Å F…F contacts. In contrast, both HS variants of the second molecule are characterized by strong close contacts, with either a pair of 2.973 Å C…F and a pair of 2.848 Å F…F contacts or a pair of 2.681 Å F…F contacts. The latter is the strongest close contact observed among all the [Pd(CH_3_CXCHCO(R))_2_] complexes. Both variants also have a 3.061 Å C…F contact on the side of the molecule, which is the same contact as in the HS of the first molecule. In contrast, the HS of polymorph **5_1** does not show strong close contacts; the only close contacts observed are two pairs of weak 2.879 Å F…F and 3.147 Å C…O contacts.

The HSs for Pd β-iminoketonates, mapped with the d_norm_ property (range −0.18 to 1.2), are shown in Figure 6. Unlike its β-diketonate analogue, the HS of **2** shows a pair of strong 2.360 Å F…H close contacts. Complex **4** contains two symmetrically independent molecules in the unit cell (№1 and №2). The front surface of the first molecule of **4** shows one relatively strong 2.416 Å F…H contact between the –C_2_F_5_ group of molecule №1 and the hydrogen atom belonging to the imino group of molecule №2. Two pairs of weak 2.893 Å F…F and 2.505 Å F…H contacts between the №1 molecules from adjacent stacks are also present on the sides of the molecule’s HS. Apart from the aforementioned strong F…H contact, the HS of molecule №2 also shows a symmetrical pair of 2.438 Å F…H contacts between the №2 molecules from adjacent stacks on its side. Finally, similarly to **4**, the HS of **6** contains a pair of strong 2.379 Å F…H contacts between the –C_3_F_7_ group and the imino group.

Another useful HS mapping scheme is the shape index mapping, which visualizes weak π–π interactions between molecules. On the displayed shape index, such interactions are displayed as pairs of red and blue triangles arranged in the shape of an hourglass. Figure 7 shows the HS for all [Pd(CH_3_CXCHCO(R))_2_] complexes where such patterns were observed.

**1**, **5_2** (only one of two symmetrically independent molecules), **2**, and **6** have the same style of π–π interactions between molecules, where the metallocycles of adjacent molecules rotate by 180° relative to each other so that the Pd atom is located above the carbon atom Cγ, and the oxygen/nitrogen atoms are located above the carbon atoms. In **5_1,** the pattern of π–π interactions is slightly different: the molecules are rotated by ~155° relative to each other so that the Pd atom is located between two carbon atoms. Finally, in **4,** the molecules are arranged in straight stacks and rotated by ~23° relative to each other so that the Pd atom of one molecule is located opposite the Pd of another molecule, and both metallocycles participate in the π–π interaction. The characteristics of the π–π interactions (the angle between metallocycles, the distance between metallocycle planes, and the in-plane shift between the centroids) are shown in Figure 5. No π–π interactions were observed in **3_1**. 

### 2.4. Lattice Energy Calculations of the Complexes **1**–**6**

The total lattice energy of the crystal (E_latt_), which is the energy per molecule required to break the crystal into non-interacting molecules, can be divided into intermolecular attractive interactions, further referred to as E_vdW_ (van der Waals binding energy per molecule) and contributions resulting from changes in the conformation of the molecule between the crystal and the gas phase, E_def_ (the molecular deformation energy): E_latt_ = E_vdW_ + E_def_(1)
E_vdW_ = (E_s_/Z) − E_d_(2)
E_def_ = E_d_ − E_g_(3)
where E_s_ is the total energy of a unit cell with previously optimized positions of hydrogen atoms, Z is the number of molecules per unit cell, and E_d_ and E_g_ are the energies of the isolated molecule with the crystal geometry (deformed) and the optimized gas-phase geometry, respectively. The calculated E_latt_, E_vdW_, and E_def_ values for the crystal structures of **1**–**6** are shown in Table 5. For structures containing two or more conformers, the lattice binding energy is averaged.

A comparison of the structures of molecules optimized in the gas phase and the structures obtained from SXRD data is shown in Figure 8. Clearly, the largest part of E_def_ refers to the deformation of the fluorinated groups due to the repulsive interaction between them in the crystal. It can be seen that the E_def_ value for Pd β-iminoketonates decreases with the elongation of the fluorine chain in the ligand. However, for Pd β-diketonates, the picture is much more complicated due to polymorphism.

Due to the absence of strong directed intermolecular bonds, polymorphism is quite common in molecular crystals. While the studied Pd β-iminoketonates contain relatively strong intermolecular bonds such as as the N-H…F contacts, these are not present in the corresponding Pd β-diketonates. Hence, we were able to crystallize two polymorphs for **3** and three polymorphs for **5**. 

The DFT calculations of E_latt_, E_vdW_, and E_def_ allowed us to compare the stability of these structures. Polymorphs **3_1** and **3_2** have very similar E_vdW_ values (131.9 kJ/mol and 130.4 kJ/mol, respectively). However, polymorph **3_2** is much more strained, as indicated by its E_def_ value (56.9 kJ/mol vs. 5.3 kJ/mol for the **3_1**). As a result, the polymorph **3_1** is 53.4 kJ/mol more stable than **3_2**. This conclusion is confirmed by the analysis of the XRD experiments (Figure 8). Comparison of the polymorphs **5_1** and **5_2** shows that the latter, although it has slightly lower deformation energy, has a significantly lower E_vdW_, which makes it less stable. The polymorph **5_3** has both a lower E_vdW_ and a higher E_def_ value, which makes it the least stable among the three. 

### 2.5. Thermal Properties of the Complexes **1**–**6**

The TG curves of **1**–**6** show one-step mass loss (Figure 9). The final mass of residues did not exceed 6%. This indicates the excellent thermal stability of **1**–**6**, making them all suitable precursors for CVD/ALD applications. The onset temperatures of mass loss for all complexes occurred within a small interval (95–110 °C). Note that the mass losses of both **5_1** (*m.p.* 86 °C) and **5_2** (*m.p.* 94 °C) polymorphs occurred in the range of 95–200 °C, indicating that the evaporation process occurs in the same range. There was no significant difference between the TG curves of **5_1** and **5_2**. The mass losses of **1, 3_1, 4,** and **6** occurred in the range of 100–230 °C corresponding to both sublimation and evaporation processes, and only **2** (*m.p.* 214 °C) sublimed under the TG conditions. 

It should be noted that the enthalpy of sublimation ∆_sub_H correlates with the value of the lattice energy E_latt_ and can be estimated from the results of DFT calculations according to Equation (4) [34]:∆_sub_H = −E_latt_ − 2R(4)

Examples of the use of DFT calculations for the estimation of the volatility of fluorinated metal complexes with various combinations of CF_3_ groups in ligands are known [21,35,36]. An increase in volatility in a series of Pd diketonates ([Pd(CH_3_COCHCOCH_3_)_2_] vs. [Pd(tfa)_2_] (complex 1) vs. [Pd(hfa)_2_]) is interpreted, first of all, by their structural features (weakening of intermolecular interactions between neighboring stacks of molecules), leading to a decrease in the values of E_latt_(|∆_sub_H|) with an increase in the number of fluorinated groups in ligands [37].

We assume that the volatility of complexes **1**–**6** will also depend on features of the packing of molecules in crystals. However, the comparison of E_latt_ values for **1**–**6** given in Table 5 (column 4) does not reveal clear correlations between the volatility and the length of the fluorinated chain in ligands. In general, we observed an increase in volatility with an increase in the length of the fluorinated chain, but this effect seems to fade away (the effect levels off with an increase in the length of the chain). In both series of Pd β-diketonates (**1**, **2**) and the series of Pd β-iminoketonates (**3_1**, **4**), the replacement of a substituent from CF_3_ to C_2_F_5_ was accompanied by an increase in volatility from **1**, **2** to **3_1**, **4**, respectively. This effect may be due to structural features since, during the transition from **1** to **3_1**, an increase in the distances between neighboring molecules in stacks was observed (Table 4), with a change in intermolecular contacts from H…F to F…F, respectively. In the case of Pd β-iminoketonates, a weakening of the π…π interaction between metallocycles was observed during the transition from **2** to its closest analogue **4**, as well as the appearance of specific NH…F contacts contributing to the E_latt_ values (Table 5, column 4).

The subsequent increase in the length of the fluorinated chain from C_2_F_5_ (complexes **3_1** and **4**) to C_3_F_7_ (complexes **5_2** and **6**) did not significantly affect the types of intermolecular contacts and their lengths. It is known that the deformation of molecules in crystals of complexes can affect their volatility [27]. According to the DFT data, a longer fluorinated chain in ligands provides greater flexibility in the geometry of the complex in the crystal, and consequently leads to a decrease in the deformation energy, thus minimizing the differences in volatility between complexes **3, 4** and **5, 6**. (Table 5). This phenomenon was observed for the Pd β-iminoketonates (**2** vs. **4** vs. **6**), but to a lesser extent for the Pd β-diketonates (**1** vs. **3_1** vs. **5_2**). 

Comparing the volatility of complexes 1–6 depending on the combination of donor atoms in the ligand, it can be concluded that Pd β-iminoketonates (complexes 2, 4, and 6) are more volatile than their Pd β-diketonate analogues (complexes 1, 3_1, and 5_2). However, as can be seen from Figure 9, the differences in volatility between Pd β-iminoketonates and Pd β-diketonates were minimized with further elongation of the fluorine chain in the ligands. 

The observed effect may be due to the minimization of the differences between the E_latt_ values of the corresponding complexes (Table 5, column 4) as the length of the fluorinated chain increased: (1 (128.7 kJ/mol)/2(98.2 kJ/mol)) vs. (3_1 (126.7 kJ/mol)/4 (121.2 kJ/mol)) vs. (5_2 (132.6 kJ/mol)/6 (127.6 kJ/mol)). 

## 3. Materials and Methods

The palladium complexes were prepared by the standard procedure described elsewhere [38], using the following commercially available reagents: palladium acetate (Pd(OAc)_2_, CAS 3375-31-3), sodium hydride (NaH, CAS 7646-69-7), aqueous ammonia, ligands (Htfa, CAS 367-57-7), (Hpfpa, CAS 356-40-1), (Hhfba, CAS 356-30-9), and solvents. A mixture of HL_F_ (0.003 mol) and NaH (0.07 g, 0.003 mol) in methanol (10 mL) was added to a stirring solution of Pd(OAc)_2_ (0.22 g, 0.001 mol) in methanol (10 mL), giving a yellow solution. After evaporation of methanol in air, the crude products were purified by sublimation in vacuum (10^−2^ Torr, 60–130 °C). The data for the elemental analysis and NMR spectra of complexes **1**–**6** are presented below.

Complex **1** was isolated with a 75% yield, *m.p.* > 215 °C. For C_10_H_8_F_6_O_4_Pd, calculated %: C 29.1; H 1.95; F 27.6; found %: C 31.1; H 1.8; F 28.3; ^1^H NMR (500.13 MHz, CDCl_3_, δ, ppm): 5.94 (s, –CγH, 1H^cis+trans^, *^1^J*_C–H_ = 165 Hz), 2.29 (s, –CH_3_, 3H^cis^, *^1^J*_C–H_ = 130 Hz), 2.28 (s, –CH_3_, 3H^trans^, *^1^J*_C–H_ = 130 Hz); ^13^C NMR (125.76 MHz, CDCl_3_, δ, ppm): 195.5 (s, –CO^trans^), 195.0 (s, –CO^cis^), 168.7 (q, –CO^cis^, *^2^J*_C–F_ = 35 Hz), 168.1 (q, –CO^trans^, *^2^J*_C–F_ = 35 Hz), 115.9 (q, –CF_3_^trans^, *^1^J*_C–F_ = 285 Hz), 115.8 (q, –CF_3_^cis^, *^1^J*_C–F_ = 285 Hz), 97.90 (s, –CγH^trans^), 97.86 (s, –CγH^cis^), 26.77(s, –CH_3_^trans^), 26.73 (s, –CH_3_^cis^).

Complex **2** was isolated with a 71% yield, *m.p.* 214 °C. For C_10_H_8_F_6_O_2_N_2_Pd, calculated %: C 29,4; H 1,97; F 27,9; found %: C 28,7; H 1,9; F 29,1, ^1^H NMR (500.13 MHz, CDCl_3_, δ, ppm): 7.82(s, –NH^cis^, 1H^cis^), 7.34 (s, –NH^trans^, 1H^trans^), 5.50(s, –CγH^cis^, 1H^cis^, *^4^J*_H–H_ = 2.5 Hz), 5.45 (s, –CγH^trans^, 1H^trans^, *^1^J*_C–H_ = 163 Hz, *^4^J*_H–H_ = 2.5 Hz), 2.24(s, –CH_3_^cis^, 3H^cis^, *^1^J*_C–H_ = 125 Hz, *^4^J*_H–H_ = 1.1 Hz), 2.18 (s, –CH_3_^trans^, 3H^trans^, *^1^J*_C–H_ = 129 Hz, *^4^J*_H–H_ = 1.1 Hz); ^13^C NMR (125.76 MHz, CDCl_3_, δ, ppm): 167.7 (s, –CN), 159.9 (q, –CO, *^2^J*_C–F_ = 32 Hz), 118.0 (q, –CF_3_, *^1^J*_C–F_ = 281 Hz), 95.3 (s, –CγH), 26.3 (s, –CH_3_).

Complex **3** was isolated with a 79% yield, *m.p.* 162 °C. For C_12_H_8_F_10_O_4_Pd, calculated %: C 28.12; H 1.57; F 37.06; found %: C 26.4; H 1.2; F 37.3. ^1^H NMR (500.13 MHz, CDCl_3_, δ, ppm): 5.97(s, –CγH^cis^, 1H^cis^, *^1^J*_C–H_ = 165 Hz), 5.96 (s, –CγH^trans^, 1H^trans^, *^1^J*_C–H_ = 165 Hz), 2.29(s, –CH_3_^cis^, 3H^cis*1*^*J*_C–H_ = 129 Hz), 2.28 (s, –CH_3_^trans^, 3H^trans^, *^1^J*_C–H_ = 130 Hz); ^13^C NMR (125.76 MHz, CDCl_3_, δ, ppm): 195.1(s, –CO^trans^), 194.6 (s, –CO^cis^), 169.7(t, –CO^cis^, *^2^J*_C–F_ = 24 Hz), 168.8 (t, –CO^trans^, *^2^J*_C–F_ = 24 Hz), 118.0 (m, –CF_3_, *^1^J*_C–F_ = 266 Hz, *^2^J*_C–F_ = 39 Hz), 107.2 (m, –CF_2_–, *^1^J*_C–F_ = 264 Hz, *^2^J*_C–F_ = 38 Hz), 99.1(s, –CγH^trans^), 98.9 (s, –CγH^cis^), 26.82(s, –CH_3_^trans^), 26.80 (s, –CH_3_^cis^).

Complex **4** was isolated with a 75% yield, *m.p.* 147 °C. For C_12_H_8_F_10_O_2_N_2_Pd, calculated %: C 28.34; H 1.59; F 37.35; found %: C 28.9; H 1.7; F 35.7. ^1^H NMR (500.13 MHz, CDCl_3_, δ, ppm): 7.27 (s, –NH, 1H), 5.48 (s, –CγH, 1H, *^1^J*_C–H_ = 164 Hz), 2.19 (s, –CH_3_, 3H, *^1^J*_C–H_ = 129 Hz); ^13^C NMR (125.76 MHz, CDCl_3_, δ, ppm): 167.5 (s, –CN), 160.4 (t, –CO, *^2^J*_C–F_ = 23 Hz), 118.6 (m, –CF_3_, *^1^J*_C–F_ = 289 Hz, *^2^J*_C–F_ = 37 Hz), 108.5 (m, –CF_2_–, *^1^J*_C–F_ = 264 Hz, *^2^J*_C–F_ = 38 Hz), 96.6 (s, –CγH), 26.4 (s, –CH_3_).

Complex **5** was isolated with a 70% yield. For C_14_H_10_F_14_N_2_O_2_Pd, calculated %: C 28.54; H 1.65; F 43.56; N 4.58; found %: C 28.5; H 2.2; F 41.4; N 5.3 ^1^H NMR (500.13 MHz, CDCl_3_, δ, ppm): 5.96(s, –CγH^cis^, 1H^cis^, *^1^J*_C–H_ = 165 Hz), 5.94 (s, –CγH^trans^, 1H^trans^, *^1^J*_C–H_ = 165 Hz), 2.29(s, –CH_3_^cis^, 3H^cis^, *^1^J*_C–H_ = 129 Hz), 2.28 (s, –CH_3_^trans^, 3H^trans^, *^1^J*_C–H_ = 129 Hz); ^13^C NMR (125.76 MHz, CDCl_3_, δ, ppm): 195.1(s, –CO^trans^), 194.7 (s, –CO^cis^), 169.5(t, –CO^cis^, *^2^J*_C–F_ = 23 Hz), 168.6 (t, –CO^trans^, *^2^J*_C–F_ = 23 Hz), 117.5 (m, –CF_3_), 108.8 (m, –CF_2_–), 99.4 (s, –CγH^trans^), 99.2 (s, –CγH^cis^), 26.89(s, –CH_3_^trans^), 26.87 (s, –CH_3_^cis^).

Complex **6** was isolated with a 80% yield, *m.p.* 166 °C. For C_14_H_10_F_14_N_2_O_2_Pd, calculated %: C 27,54; H 1,65; F 43,56; N 4,59; found %: C 28,1; H 1,6; F 42,8; N 4,2. ^1^H NMR (500.13 MHz, CDCl_3_, δ, ppm): 7.26 (s, –NH, 1H), 5.46 (s, –CγH, 1H, *^1^J*_C–H_ = 164 Hz), 2.20 (s, –CH_3_, 3H, *^1^J*_C–H_ = 128 Hz); ^13^C NMR (125.76 MHz, CDCl_3_, δ, ppm): 167.5 (s, –CN), 160.3 (t, –CO, *^2^J*_C–F_ = 23 Hz), 117.8 (m, –CF_3_, *^1^J*_C–F_ = 288 Hz, *^2^J*_C–F_ = 34 Hz), 110.0 (m, –CF_2_–C_2_F_5_, *^1^J*_C–F_ = 262 Hz, *^2^J*_C–F_ = 30 Hz), 108.7 (m, CF_2_–CF_2_–CF_3_), 96.9 (s, –CγH), 26.5 (s, –CH_3_).

The elemental analysis (C, H, N, F) was carried out at the Vorozhtsov Novosibirsk Institute of Organic Chemistry SB RAS, using original methods [39]. The standard uncertainties of the C, H, and F determinations did not exceed 0.5%. ^1^H, ^13^C NMR spectra in CDCl_3_ were recorded on a Bruker Avance III 500 MHz spectrometer at 25 °C. ^13^C and ^1^H shifts were referenced to the external tetramethylsilane standard. The infrared spectra of **1**–**6** were recorded with a VERTEX 80 FTIR spectrometer.

SXRD analysis of crystals of **2**–**6** and polymorphs of **3** and **5** was carried out using Bruker X8 (MoKa sealed tube with graphite monochromator, APEX II CCD detector, 4-circle kappa goniometer) and Bruker D8 Venture (MoKa Incoatec IμS 3.0 microfocus X-ray source, PHOTON III CPAD detector, 3-circle kappa goniometer) single-crystal diffractometers. In both cases, the sample temperature was maintained at a constant value using an Oxford Cryosystems Cryostream 800 plus open-flow nitrogen cooler. The data collection, reduction, absorption correction, and unit cell refinement were performed using the APEX3 v2018.7-2 software package (SAINT 8.38A, SADABS-2016/2). The resulting *hkl* datasets were processed with Olex2 v.1.2.10 [40], using SHELXT 2018/2 [41] and SHELXL 2018/3 [42] for the structure solution and refinement, respectively. 

Powder XRD patterns of **1**–**6** were recorded using a Shimadzu XRD-7000 powder diffractometer (CuKa sealed tube with Ni β-filter, Bragg–Brentano geometry with a vertical θ–θ goniometer, 0.0143° scan step, and OneSight silicon strip array detector). Samples were prepared as thin powder layers on the surface of a glass cuvette. The patterns were taken in step-by-step mode in the angular range 2θ = 2–35°.

Hirshfeld surface (HS) analysis was performed in CrystalExplorer 17.5 [43] with a Tonto computational chemistry package [44].

DFT calculations were performed in BAND2014 [45], and gas-phase calculations were performed in ADF2017 [46]. Both programs use specific basis sets composed of Slater-type functions [47]. The PBE-D3(BJ) [48,49] level of theory was used. Scalar relativistic effects were included via zeroth-order regular approximation (ZORA) [50]. Optimization of hydrogen atom positions in the crystal structure was performed using a double-ζ quality basis with a set of polarization functions (DZP). The 1s orbitals of F, O, and C atoms and the lower orbitals of Pd atoms (up to 3d) were frozen. After that, single-point calculations for the crystals were performed with all-electron basis sets of TZ2P quality for all atoms. For single-molecule (gas phase) calculations, all-electron basis sets of TZ2P quality were also used for all atoms.

TG measurements were carried out in a helium atmosphere (30 cm^3^·min^−1^) using Netzsch STA 449 C thermo-analytical equipment (Al_2_O_3_ crucible, with a heating rate of 10 °C∙min^−1^, a temperature interval of 25–250 °C, and a sample mass of 5.0 ± 0.5 mg).

## 4. Conclusions

Series of Pd β-diketonates [Pd(CH_3_COCHCO(R))_2_] and β-iminoketonates [Pd(CH_3_CNHCHCO(R))_2_], with different lengths of the fluorinated substituent chain (R = C*_n_*F_2*n*+1_) in ligands, have been studied in depth. These compounds, including new ones with substituents R = C_2_F_5_ and C_3_F_7_ in the ligands, were synthesized with yields of 70–80% and characterized by a set of experimental (SXRD, XRD, IR, NMR spectroscopy, TG) and theoretical (DFT, Hirshfeld surface analysis) methods. NMR studies showed that both *cis* and *trans* isomers existed in solutions of Pd β-diketonates, but only *trans* isomers were observed in the case of Pd β-iminoketonates.

The peculiarity of [Pd(CH_3_COCHCO(R))_2_] with the substituents R = C_2_F_5_, C_3_F_7_ is the presence of several structural polymorphs. A systematic structural study of the series of palladium complexes, including their polymorphs, indicated that flat molecules of the complexes tend to assemble in stacks and to be arranged either uniformly or in a herringbone pattern. 

This packing style was distorted with the introduction of R = C_2_F_5_, C_3_F_7_ substituents in ligands, and the molecules in [Pd(CH_3_COCHCO(C_2_F_5_))_2_] crystals were packed in layers rather than stacks. According to the Hirshfeld surface analysis, the main contacts involved in the formation of stacks of Pd complexes with the substituents R = CF_3_ were H…F contacts, while C…F, F…F, and NH…F contacts appeared in the structures of complexes with elongated fluorine chains in the ligands. The DFT studies of the stability of the complexes and their polymorphs in the crystal phases correlated with the XRD data, and two polymorphs of [Pd(CH_3_COCHCO(C_3_F_7_))_2_] were isolated as individual phases.

In the series of Pd β-diketonates and Pd β-iminoketonates, the volatility increased with the elongation of fluorinated chains in the ligands only from CF_3_ to C_2_F_5_ substituents. A longer fluorinated chain in the ligands provided greater flexibility in the geometry of the complex in the crystal, and consequently led to a decrease in the deformation energy, minimizing the differences in volatility between the complexes with substituents C_2_F_5_ and C_3_F_7_. The difference in volatility between Pd β-iminoketonates and Pd β-diketonates was minimized with further elongation of the fluorine chains in the ligands. All complexes exhibited high volatility and thermal stability, making them promising candidates for CVD/ALD applications.

## Data Availability

The data presented in this study are available on request from the corresponding author. The CIF files containing full crystallographic data for **2, 3_1, 3_2, 5_2, 5_3, 5_1, 4,** and **6** (CCDC 2129497–2129504) can be obtained free of charge via http://www.ccdc.cam.ac.uk/conts/retrieving.html, accessed on 4 March 2022 (or from the CCDC, 12 Union Road, Cambridge CB2 1EZ, UK; Fax: +44-1223-336033; Email: deposit@ccdc.cam.ac.uk).

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
