# Peer review of "The Theoretical and Experimental Investigation of the Fluorinated Palladium β-Diketonate Derivatives: Structure and Physicochemical Properties"

_molecules, 2022, doi:10.3390/molecules27072207_

Round 1

Reviewer 1 Report

Svetlana I. Dorovskikh et al reported The theoretical and experimental investigation of the fluorinated palladium β-diketonate derivatives: structure and physicochemical properties. Series of Pd β-diketonates [Pd(CH3COCHCO(R))2] and β-iminoketonates [Pd(CH3CNCHCO(R))2] with different lengths of the fluorinated substituent chain (R = CnF2n+1) in ligands has been studied, In my point of view it is acceptable after major revision.

1- Introduction must rewritten again as it is very large.

2-  English language and style are major spell check required.

3- For 1, 3 and 5 each signal in both 1H and 13C spectra is doubled, which indicates that,most likely, the solutions contain both cis and trans isomers of the complexes. In principle, such isomers are possible for all complexes under study. How possible for all if signals are not doubled for all? onle 1,3,5 are doubled!

4- This agrees with the fact that solutions of 4 and 6 contain only one isomer. As mention before fron author that that all complexes have 2 isomers. Which is correct?

5- The 1H NMR spectrum of the solution of 2 contains an additional set of small signals, which, most likely, relate to a small admixture of the cis isomer. Explain this?

6- In Figure 7 Why distance and shift are ratio and different from other complexes?

Author Response

We would like to acknowledge the Reviewer for careful reading and analysis of our manuscript. Below our replies and comments on all specific points indicated by the Reviewer have been presented.

Point 1: Introduction must rewritten again as it is very large.

 Response 1: The introduction was rewritten.

Point 2: English language and style are major spell check required.

Response 2: English language and style were checked.

Point 3: For 1, 3 and 5 each signal in both 1H and 13C spectra is doubled, which indicates that,most likely, the solutions contain both cis and trans isomers of the complexes. In principle, such isomers are possible for all complexes under study. How possible for all if signals are not doubled for all? onle 1,3,5 are doubled!

Response 3: We have replaced “in principle” with “theoretically” (page 3). Theoretically, for each complex two isomers are possible. However, according to DFT calculations, for 2, 4, and 6, trans isomers are much more stable than cis isomers (see Table 2 on page 4). We also provide NMR spectra of complexes in Supplementary.

Point 4: This agrees with the fact that solutions of 4 and 6 contain only one isomer. As mention before fron author that that all complexes have 2 isomers. Which is correct?

Response 4: DFT calculations predict two isomers for all complexes under study. However, for 2, 4, and 6, trans isomers are much more stable than cis isomers (see Table 2 on page 4). In our experiments (room temperature, CDCl3 solution), we do not see cis isomers of 4 and 6. Probably, cis isomers of 4 and 6 may be experimentally observed under some other conditions.

Point 5: The 1H NMR spectrum of the solution of 2 contains an additional set of small signals, which, most likely, relate to a small admixture of the cis isomer. Explain this?

Response 5: Probably, the wording is incorrect. We have replaced “relate” with “refer” (page 4).

As we mentioned before, DFT calculations predict that the cis isomer of 2 is much less stable than the trans isomer. However, in the CDCl3 solution, a small amount of 2 transforms to a cis isomer (we are sure that initially we dissolve only the trans isomer because we observe only trans isomers in the crystal structures). Cis-trans transition is probably possible because the ligands in 2 are the smallest (geometrically) and, as a result, the steric hindrance effects are smaller.

Kind regards,
Dr. Svetlana Dorovskikh

Point 6: In Figure 7 Why distance and shift are ratio and different from other complexes?

Response 6: Compound 4 is the only compound whose crystal structure contains two pairs of π-π interactions between neighboring molecules. Therefore we provided two sets of values (angle, distance and shift) separated by “/” sign, which made it look like a ratio instead of two separate values. We amended that mistake by separating one label into two and adding pointing arrows. The Fig 7 was changed.

Reviewer 2 Report

The paper is a continuation of research on CVD precursors carried out in scientific group of Tamara V. Basova. The authors describe synthesis, structural, spectral (IR, NMR), theoretical (DFT and Hirshfeld Surface intermolecular interaction analysis) and thermogravimetry studies on a series of palladium β-diketonate or β-iminoketonate derivatives, modified with three types of fluorinated alkyl chains CF3, C2F5 or C3F7.  The subject is relevant to the search for optimal precursors of CVD structurised deposits of palladium for catalytic purposes or for application in sensors. The manuscript describes six chemically distinct compounds, two of which were described previously in works referenced as [33] for compound 2 and [34] for compound 1. Four compounds are new, all were structurally characterised by X-ray diffraction analysis, compound 3 as two polymorphs and 5 in three polymorphous modifications. The authors performed a detailed characterisation of the products focusing on parameterers crucial for CVD applications as lattice energy, melting point and enthalpy of sublimation. They conclude volatility for Pd  β-iminoketonates is better than for Pd  β-diketonates, however long  C3F7 chains do not give significant improvement of volatility anymore.

I recommend also publishing retermination of structure 2, since it is of much better quality than deposited as CCDC 1224321 and contains coordinates of hydrogen atoms.  

The manuscript may be of interest for relatively broad range of readers. The data are of good quality and generally spoken adequately described, but a considerable mess is encountered in the manuscript as described below, so a careful revision is required.

Table 3. Identification code for the redermined structure in the first column should be 2 not 1! This is iminoderivative i-tfa which is consistent with the chemical formula! Please change the relevant text accordingly in lines 159-162 and ALL references below. Alternatively you may change codes definded in Table 1, but then we would loose simplicity given by correlation of numbers’ parity and substitution of diketonates.

Moreover, statement “data for 2, 3_1, 3_2, 4, 5_1, 5_2, 5_3, 6 (CCDC 2129497–2129504)” in lines 543-543 is misleading since the items contain actually compounds in another order: 2, 3_1, 3_2, 5_2, 5_3, 5_1, 4 and 6 inconsistent with numeration in the manuscript and not provided correctly here. Nevertheless, here we have structure 2 instead of 1 on the first position (see my first comment). Numbers 4 and 5_1 are put out of order, anyway. The reason may be related to the order of uploading files disturbed unintentionally during deposition, but this should/must be verified afterwards!

Minor or editorial errors

line 108 “fluorine chain in LF” LF is not defined.

line 499 “[Pd(CH3CNCHCO(R))2]” should be replaced with “[Pd(CH3CNHCHCO(R))2]” one hydrogen atom is missing.

Line 404 “as can be seen from Fig. the differences…“ no Figure number is supplied! Please indicate one, probably 9. 

Line 406 “between the Elat values” please use subscript for lattice energy Elatt.

Author Response

We would like to acknowledge the Reviewer for careful reading and analysis of our manuscript. Below our replies and comments on all specific points indicated by the Reviewer have been presented.

Point 1: I recommend also publishing retermination of structure 2, since it is of much better quality than deposited as CCDC 1224321 and contains coordinates of hydrogen atoms.  

The manuscript may be of interest for relatively broad range of readers. The data are of good quality and generally spoken adequately described, but a considerable mess is encountered in the manuscript as described below, so a careful revision is required.

Table 3. Identification code for the redermined structure in the first column should be 2 not 1! This is iminoderivative i-tfa which is consistent with the chemical formula! Please change the relevant text accordingly in lines 159-162 and ALL references below. Alternatively you may change codes definded in Table 1, but then we would loose simplicity given by correlation of numbers’ parity and substitution of diketonates.

Response 1: We agree with the reviewer's opinion. The refined structure of 2 has been deposited (CCDC 2129497). Of course, the structure of 2, not 1 is mentioned in Table 3 and the text (lines 159-162) and other references. This inaccuracy has been corrected.

Point 2: Moreover, statement “data for 23_13_245_15_25_3(CCDC 2129497–2129504)” in lines 543-543 is misleading since the items contain actually compounds in another order: 2, 3_1, 3_2, 5_2, 5_3, 5_1, 4 and 6 inconsistent with numeration in the manuscript and not provided correctly here. Nevertheless, here we have structure 2 instead of 1 on the first position (see my first comment). Numbers 4 and 5_1 are put out of order, anyway. The reason may be related to the order of uploading files disturbed unintentionally during deposition, but this should/must be verified afterwards!

Response 2: The reviewer is right, and the structure of 2 is meant here instead of 1. Indeed, there was an unfortunate confusion in the order of the files when they were uploaded. The structures of CCDC 2129497-2129504 complexes have the following correct order: 2, 3_1, 3_2, 5_2, 5_3, 5_1, 4, and 6. To avoid confusion, this order is given in the revised article.

Point 3: Minor or editorial errors

line 108 “fluorine chain in LF” LF is not defined.

line 499 “[Pd(CH3CNCHCO(R))2]” should be replaced with “[Pd(CH3CNHCHCO(R))2]” one hydrogen atom is missing.

Line 404 “as can be seen from Fig. the differences…“ no Figure number is supplied! Please indicate one, probably 9. 

Line 406 “between the Elat values” please use subscript for lattice energy Elatt.

 Response 3: These errors have been corrected.

Kind regards,
Dr. Svetlana Dorovskikh

Round 2

Reviewer 1 Report

accept